# A Quality Improvement Initiative to Reduce Postoperative Delirium among Cardiac Surgery Patients

**DOI:** 10.3390/geriatrics6040111

**Published:** 2021-11-16

**Authors:** Rohan M. Sanjanwala, Brett Hiebert, David Kent, Sandy Warren, Hilary Grocott, Rakesh C. Arora

**Affiliations:** 1Cardiac Science Program, St Boniface Hospital, Winnipeg, MB R2H 2A6, Canada; rsanjanwala@sbgh.mb.ca (R.M.S.); bhiebert3@sbgh.mb.ca (B.H.); dkent@sbgh.mb.ca (D.K.); swarren@sbgh.mb.ca (S.W.); hgrocott@sbgh.mb.ca (H.G.); 2Department of Anesthesia, Perioperative and Pain Medicine, Max Rady College of Medicine, University of Manitoba, Winnipeg, MB R3E 0W2, Canada; 3Department of Surgery, Section of Cardiac Surgery, Max Rady College of Medicine, University of Manitoba, Winnipeg, MB R3E 0W2, Canada

**Keywords:** postoperative delirium, postoperative care, quality improvement, cardiac surgery, care strategies

## Abstract

Patients following cardiac surgery commonly experience post-operative delirium (POD) during their postoperative hospital stay. A multifaceted, specialty wide, quality improvement (QI) project was undertaken for patients experiencing POD. The goal was to develop a reduction in POD care bundle (rPOD-a structured patient care program) that encompasses efficient preoperative risk factor identification and a postoperative patient-care process to ensure early POD identification and treatment. The following steps were taken to implement the rPOD care bundle including: (a) Developing a quality driven, evidence-based guideline for the perioperative cardiac surgery health care team, (b) identifying and addressing local barriers to implementation, (c) selecting performance measures to assess intervention adherence and patient outcomes, and (d) ensuring that all patients receive the interventions through staff engagement and education, and regular project evaluation. Trends of process measures and quality improvement measures were examined. An increasing trend in the rate of postoperative delirium screening during implementation of rPOD intervention was demonstrated. This quality improvement study provides a bases for future postoperative delirium reduction interventions.

## 1. Introduction

Delirium is an acute brain dysfunction characterized by a fluctuating disturbance of consciousness with inattention and cognition and perception deficits [1,2]. It is the most common neuropsychological complication following cardiac surgery, with the most recent estimates of prevalence ranging from 25% to 50% [3,4]. While postoperative delirium (POD) may resolve in hospital, the patients experiencing delirium are at a higher risk of worse in-hospital and post-discharge outcomes. Numerous studies among cardiac surgery patients have demonstrated that POD increases the risk of postoperative mortality and of prolonged length of hospital stay [5,6,7,8]. In addition, such patients are at a higher risk of functional decline both physically and mentally, resulting in post-discharge poor quality of life, as well as a higher rate of nursing home placement [5,6,7,8]. There has been extensive research documenting the short- and long-term hazards associated with POD. In the same vein, many critical care societies have published care bundles to support institutional patient-care processes facilitating early identification and treatment of POD. One such example would be of the society of critical care medicine published ABCDEF- ICU liberation 2010 and PAD (Pain, agitation and delirium) guidelines in 2013. However, the condition remains frequently unrecognized (in three out of four ICU patients), and under-appreciated during hospitalization [9,10].

Hospital-wide strategies focused on reducing POD are essential to improving clinical outcomes of surgery, as well as to improving patient-related outcomes including postoperative cognitive functioning. Through this knowledge translation-quality improvement initiative, the goal was to develop steps essential to implement an rPOD care bundle, centered around assessing baseline vulnerability (preoperative risk factor assessment), implementing preventative strategies, as well as early identification and management of delirium based on the available best practice evidence.

This project was implemented through a validated QI model (Figure 1) [11]. Here, we describe our collaborative healthcare improvement initiative carried through the following four steps: (1) summarizing evidence to identify potentially beneficial interventions, (2) identifying local barriers to implementation, (3) selection and development of performance measures and, (4) ensuring that all patients receive the interventions. The last step follows an iterative “4E” algorithm to engage and educate front line staff, execute the intervention and evaluate performance using objective measurement tools (Figure 1) [11].

The purpose of this quality improvement project was to develop and implement a structured program to assess risk factors and to reduce incidence, early identification and prompt treatment of POD among the post-cardiac surgery ICU patients. This project targeted health care providers in the cardiac pre-assessment clinic (CPAC), cardiac surgery in-patient unit (CSIU), intensive care cardiac surgery (ICCS), cardiology inpatient unit, inclusive of nursing staff, physician assistants, physicians, as well as cardiac surgery patients and their caregivers.

The global aim of this quality improvement project was to decrease the prevalence of postoperative delirium among cardiac surgery patients at our center. The objective of this paper is to describe a multifaceted, quality improvement undertaking for the reduction of postoperative delirium, via prevention and management, in a cardiac surgery unit within a tertiary care hospital. In doing so, the study team discusses barriers encountered during the QI process and solutions to those barriers, along with the issues concerning staff adherence and long-term sustainability in order to provide relevant information for other cardiac surgery units that may wish to undertake a similar project.

## 2. Methods

### 2.1. Context for the QI Project

The rPOD quality improvement initiative was carried out at the St. Boniface Hospital (tertiary care center for the province of Manitoba, Cardiac Science Program including CPAC, ICCS, and CSIU). The multidisciplinary delirium working group, the perioperative health care providers, Winnipeg Regional Health Authority (WRHA), and cardiovascular surgery patients were the key stakeholders.

### 2.2. Use of an Established QI Model

The rPOD care bundle was implemented starting in 2012 with its completion in 2016. Here, we have employed an established QI model (Figure 1, Table 1) for improving the quality of care for cardiac surgery patients and to reduce the postoperative delirium prevalence among the cardiac surgery patients. The strategies were employed across the time frame, which was consecutive, but at times simultaneous.

### 2.3. Applying the QI Model to Reduce Postoperative Delirium among the Cardiac Surgery Patients

Overall Considerations

The improvement process involved a large patient care system involving an extensive multidisciplinary collaboration. A key first step was establishing a multidisciplinary team (i.e., delirium working group) to design and implement the project. This process was initiated by a cardiac surgeon (RA) with extensive QI experience and who is the director of ICCS, as well as the section head for the section of cardiac surgery. The other members of the delirium working group included an anesthetist (HG), a nurse champion (SW), as well as the perioperative cardiac surgery nursing staff.

**Step 1:** Summarize the Evidence

Our QI team developed guidelines targeted towards reducing postoperative delirium. The rPOD care bundle was developed based on our previous experience, the best evidence from the literature and through expert consensus. The rPOD care bundle consists of 3 domains as follows (Table 2):

**Domain 1**: Establishing Assessment Practices

(A) The baseline vulnerability assessment, including frailty assessment (i.e., Clinical Frailty Scale (CFS)), and cognitive assessment (i.e., Montreal Cognitive Assessment (MoCA)), were implemented. These screening tools were a part of the preoperative assessment package (initially introduce by the Winnipeg Regional Health Authority as a part of a delirium program for all surgical specialties). In addition, a delirium score card (Appendix A) was implemented to assess the risk of postoperative delirium.

(B) During the postoperative period, the Confusion Assessment Method for the ICU (CAM-ICU) screening, [12] and the Critical-Care Pain Observation Tool (CPOT) [13], were introduced in the ICCS unit.

**Domain 2**: Introducing and Implementing Preventative Strategies

(A) Preoperative: The patient and caregiver delirium education brochure was given during their preoperative visit. The goal was to increase patients’ and caregivers’ awareness regarding postoperative delirium (i.e., risk factors, signs and symptoms and outcomes). In addition, the brochure provided specific guidelines regarding the patients’ and caregivers’ roles in aiding early identification and management of delirium.

A ‘Getting to know you’ form was introduced to gather personal patient information, including family members’ names, use of assistive devices (hearing aid, glasses, dentures, mobility aid), profession, interests/hobbies and preferred terms for common activities. Such information could be valuable to the frontline staff to facilitate the cognitive functions during the postoperative recovery period.

(B) Intra operative: A delirium score card was included in the operating room time-out. The patients identified as high risk of delirium, using the delirium score card, received an EEG and cerebral capnography-guided anesthesia, to minimize sedation-related neurocognitive dysfunction.

(C) Postoperative: Early mobilization protocol was introduced in the postoperative intensive care unit.

**Domain 3:** Delirium Care Strategies

The delirium care strategies included investigating modifiable factors, non-pharmacological and pharmacological interventions (Table 3). In addition, the primary care practitioners were informed of their patients’ experiencing delirium and were sent an information pamphlet describing the long-term impact of postoperative delirium which may affect their patient’s post-discharge health-related quality of life.

**Step 2:** Identify Local Barriers to Implementation

We carefully considered the steps involved in preparing the frontline staff through engaging all relevant stakeholders (ICCS, CSIU, CPAC RN, Physician Assistant) in monthly, multidisciplinary, delirium working group meetings. The goal was to identify barriers to achieve staff buy-ins and implementation. As described in step 4: multiple strategies of the 4 E’s model were applied to promote efficient implementation.

The barriers to the rPOD QI project and their management strategies are described in Table 4. An important barrier was the lack of cardiac surgery-specific delirium management guidelines. Through the Plan-Do-Check-Adjust iterative cycle the delirium working group drafted the cardiac science delirium guidelines.

The majority of local barriers were regarding intervention implementation. The rPOD intervention was implemented in sequential and additive stages to limit overwhelming the staff with multiple interventions. Lack of delirium-related education and training among the cardiac surgery frontline health care providers was another important barrier. Multiple information and education sessions were conducted to facilitate staff buy-in. The training was further augmented through the regional health authority’s initiative to educate and train all surgical units and patient care staff regarding postoperative delirium and to provide necessary training including assessment tools (CAM ICU, CPOT) and recoding in the electronic patient record.

**Step 3:** Performance Measure

The process and outcome measures were collected retrospectively (Table 5). Adherence data includes completion rates of baseline risk and POD assessment.

The primary outcome measure was the change in the rates of POD screening before, during and after the rPOD care bundle implementation. The quality indicator measures are detailed in Table 5. The performance measures were collected for the ICCS unit, Preoperative assessment clinic and CSIU. In addition, fiscal trends, in terms of departmental expenditure, constant care resource utilization for the duration of rPOD implementation and post implementation were also collected.

For the primary analysis, a pre-post design was used to compare outcomes during the baseline pre-intervention period versus when all the rPOD care bundle had been implemented.

Research ethics board approval was provided to collect compliance and quality improvement metrics data for this study

**Step 4:** Ensuring Complete Intervention Implementation: 4 Es’ Model

**Engage:** Engagement of all stakeholders, from the study leadership to frontline clinical staff, was necessary to ensure buy-in and sustained project adherence. The engagement process included: (1) Conducting educational sessions with a focus on evidence regarding risk factors of delirium, significance of incident delirium and care strategies, as well as presenting results from the previous research delirium study demonstrating delirium rates among cardiac surgery patients at our center, (2) Recruitment of a nurse champion to collaborate on the project, and (3) Monthly review meetings of the delirium working group committee.

**Educate:** Staff education took place throughout the project. To engage the frontline staff, we administered multiple education and presentation sessions. In these sessions we presented the results of our delirium screening research study and also trained staff to implement the delirium intervention. The nurse champion attended multidisciplinary team meetings and was instrumental in the project design. In preparation for the pilot study, ICCS nurses were briefed on details of the QI project, the daily checklist and delirium screening. During the pilot delirium research study, a member of the ICCS delirium team met frequently with the night shift staff to provide feedback, answer questions and address barriers to future interventions and assessment (baseline risk and delirium screening) completion. In addition, the nursing staff attended educational programs developed by the regional health authority.

**Execute:** As described in the QI model, there are four general approaches to overcoming implementation barriers. First, the QI team standardized care by orienting all intensive care staff to rPOD care bundles. Second, the QI team used independent reminders, in the form of a checklist and daily verbal reminders from charge nurses and nurse champions to complete the rPOD intervention. Additionally, the QI team introduced a smaller number of interventions at one time using a staged approach. Finally, to learn from problems, throughout the project barriers were reviewed and addressed at monthly delirium working group meetings (Step 2: Identify Local Barriers to implementation).

**Evaluate:** An “audit and feedback” approach was employed to access group adherence as well as to encourage following of the set standards. The adherence data was presented and discussed with the nurse champion during the monthly delirium working group. Those with high adherence were commended while those with lower adherence were further engaged and educated to facilitate compliance.

## 3. Results

The delirium quality improvement intervention was implemented from 2012 to 2016. Different domains of the intervention may have been implemented simultaneously (Appendix A). The adult patients (age ≥ 18 years) undergoing cardiac surgery were eligible for the intervention and outcome analysis. The corresponding health care providers’ compliance data was collected during the cardiac surgery patients’ perioperative care transition, including the preoperative, intraoperative and postoperative period. The data about each reporting period has been collected and analyzed.

The median patient age was 67 years (58–74) Table 6. The preoperative risk, as assessed using MoCA, CFS and PHQ9 (Table 6), were similar between patients across all intervention stages. Additional baseline and ICU data are summarized in the Table 7 and Table 8.

A total of 3340 patients were evaluated over the study period. We compared the unadjusted assessment completion rates across the intervention period (Table 7). The mean daily preoperative MoCA delivery and completion rates improved from 30% (30.7% for CFS) during the r-POD implementation phase to 46.6% (49.2% for CFS) in the post implementation phase. However, postoperative delirium assessment rates (considering data of the fully- or partially-completed delirium assessment tool) were similar to rates during and after the post implementation phase.

The quality improvement metrics (outcome-data) before, during and after the implementation of the rPOD intervention are reported in Table 8. The primary outcome measure for the rates of postoperative delirium screening (data only from completed CAM assessments were included) increased during the implementation of the rPOD care bundle and following its implementation. Figure 2 demonstrates the trends of gross expenditure (GL dollars) (Panel A), average patient day constant care or close observation hours and expenditure (Panel B and Panel C), and average in-patient-days (Panel D).

## 4. Discussion

In this study, the QI team used an established QI model to implement the rPOD intervention across the various departments. This model employed a previously successful 4E’s algorithm (engage, educate, execute, and evaluate). Essential to this effort was the implementation of the rPOD care bundle, which included the preoperative baseline assessment, perioperative preventative strategies and postoperative rapid delirium assessment and care strategies, in successive stages to allow for incremental adoption of the intervention. Using this approach, the QI team demonstrated that multidisciplinary, perioperative and patient-focused interventions for the reduction of postoperative delirium were feasible to be performed on a daily basis, assisting the perioperative care process.

The sustainability of QI projects are challenging. Sustainability is supported by immediate, visible results, which can be difficult in delirium-related projects. Furthermore, identifying and measuring the implementation of new initiatives within health care is difficult without sustainability of these programs over time. Continued staff education delineating the consequences of delirium, frequent feedback to support intervention adoption and identifying patient perspectives could help with the adherence for such an intervention.

A limitation to this project was that more patients underwent CABG during the initial period of this study. Over time more complex surgeries including single and double valve replacement surgery were conducted. This change could have resulted in different levels of cardiac surgical stress and in turn, could have affected the rates of delirium as well as the level of care provided to the patients. Another limitation of this project was, as with other QI projects, uncertainty regarding the generalizability of these results. This project was implemented in a setting led by academic and clinical experts and team members with training and experience in QI projects. However, many of the implementation challenges surmounted in this project, such as continued use of the intervention as part of routine care, are universal to all cardiac surgery units. Furthermore, the established QI model used and the commonsense appeal of these evidence-based rPOD interventions may facilitate its utilization and buy-in for other settings.

## 5. Conclusions

Using an established QI model to implement a multifaceted rPOD intervention to improve delirium care strategies is feasible. The future direction includes the development of strategies to address sustainability and the extension of similar efforts to other cardiac and non-cardiac surgical programs.

## Figures and Tables

**Figure 1 geriatrics-06-00111-f001:**
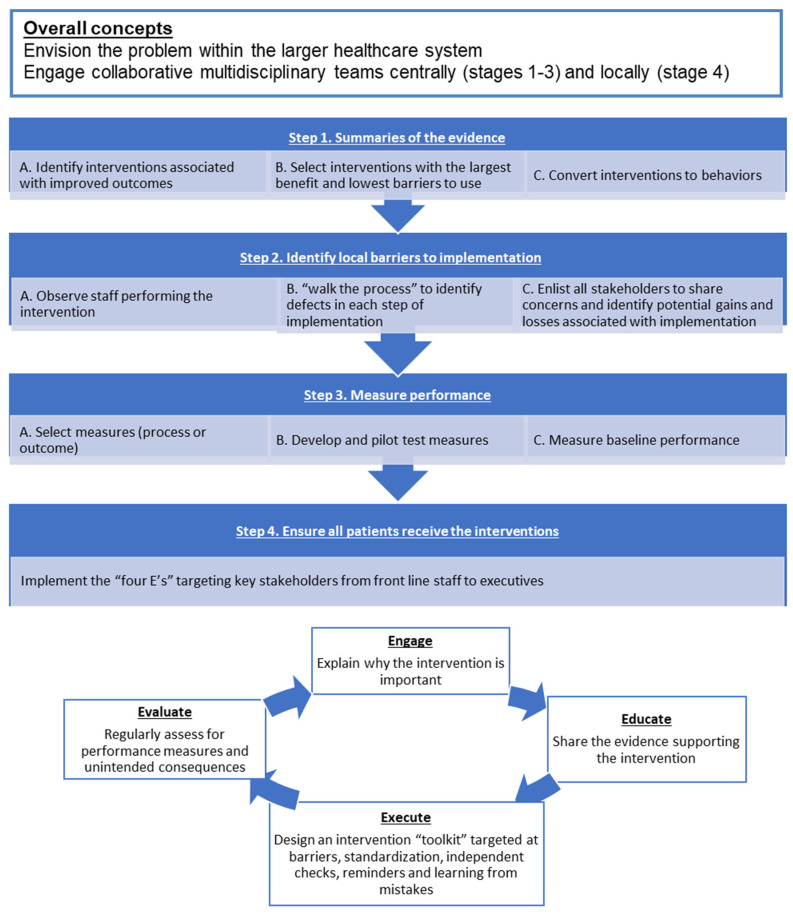
A model to implement quality improvement intervention in a healthcare setting.

**Figure 2 geriatrics-06-00111-f002:**
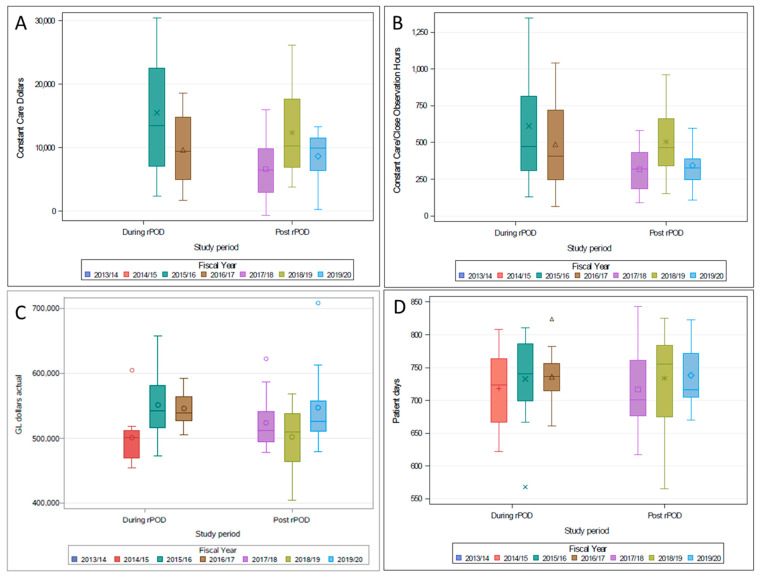
The fiscal trends during (2013–2016) and following (2017–2020) rPOD intervention. Panel (**A**)—Constant care expenditure by study period, Panel (**B**)—Constant care observation by study period. Panel (**C**)—Gross expenditure by study period and Panel (**D**)—In-patient days by study period. The financial data was collected from the administrative database with the cardiac science program.

**Table 1 geriatrics-06-00111-t001:** Quality improvement model.

Step 1: Summarizing Evidence to Identify Potentially Beneficial Interventions	Step 2: Identifying Local Barriers to Implementation	Step 3: Selecting and Developing Performance Measures	Step 4: Ensuring All Patients Receive the Interventions
1. Feasibility clinical trial for determining delirium incidence among cardiac surgery patients − The study assessed delirium incidence among cardiac surgery patients as well as, implementation barriers (such as staff education) 2. Evidence review 3. Expert consensus	1. Lack of leadership − Establishing cardiac science multidisciplinary delirium working group 2. Identifying local barriers − Monthly multidisciplinary delirium working group meetings were organized to identify barriers achieving staff buy-in and performing specific intervention. 3. Lack of knowledge and training − A delirium education package was developed consisting of -multiple educational sessions regarding the rational and evidence for rPOD care bundle as well as training RN and PA regarding assessing postoperative delirium incidence as well as preoperative baseline delirium risk − Implementation of yearly mandatory delirium self-learning module	1. Compliance measures Baseline risk assessment (A) Preoperative − Montreal Cognitive Assessment (MoCA) − Clinical Frailty Scale (CFS) (B) Postoperative − CAM-ICU − RASS 2. Performance measures (A) Primary outcome − Postoperative delirium screening rate (B) Other quality indicator	E-Engage E-Educate E-Execute E-Evaluate

The table describes the adaption of a previously validated quality improvement model and provides a stepwise implementation of the rPOD care bundle.

**Table 2 geriatrics-06-00111-t002:** Components of rPOD care bundle.

Preoperative Intervention (CPAC, CSIU, 5A)—Preoperative Assessment Package	Operative Intervention	Postoperative Intervention—ICCS
Cognitive assessment—MoCA Frailty Screening—CSF “Getting to know you” form Delirium brochure patient education Family brochure-family mental health Delirium score card PCP letter for mental health	8. Timeout-Delirium score card 9. Cerebral capnography and EEG directed anesthesia for high risk cases	10. A four hourly delirium screening using Confusion Assessment Method in intensive care unit (CAM ICU) 11. CPOT pain assessment 12. Non-pharmacological intervention 13. Early mobility 14. PCP—delirium information letter 15. Delirium order set
Nurse education package and Yearly self-learning module		

The above table delineates the components of rPOD care bundle that were implemented across the perioperative period.

**Table 3 geriatrics-06-00111-t003:** Delirium care strategies: the delirium order set.

A Positive CAM Indicates Delirium, a Medical Emergency and Should Be Translated to Following Action	
Investigations	Interventions
− Vital signs and oxygen saturations − Pain assessment (CPOT) − Blood sugar − Bladder scan to rule out retention − Last bowel movement to rule out constipation − Review fluid input and output to rule out dehydration − Access sensory alterations to evaluate need for glasses, hearing aid, sleep deprivation − Review lab results, chest x-rays and EKG − Consider cultures- blood urine, sputum, wound − Consider CT head − Review medication especially (anticholinergics) − Avoid polypharmacy − Avoid benzodiazepines, consider antipsychotic for agitated delirium	− Behavior: T-A-DA T: tolerate as much as possible A: anticipate what agitates them DA: don’t agitate them − Sleep-wake cycle promotion: at night use soft voice, lights out, ear plugs, eye masks, promote comfort. − Balance rest and activity; mobilize restless patient if safe − Remember drugs is equal to unconscious, not normal restorative sleep − Cognition and Communication: Frequent orientation but do not argue or dispute delusion. Clocks, calendars, hearing aids, glasses, stimulation activities such as cards, crosswords, Sudoku, puzzles

**Table 4 geriatrics-06-00111-t004:** Barriers to implementing rPOD intervention at St. Boniface Cardiac Surgery Program.

	Barriers	Strategy to Overcome Barriers
1	Lack of leadership	Delirium committee formation consisting of non-physician staff Establishing cardiac science multidisciplinary delirium working group with scheduled monthly meetings with a goal of delirium project planning
2	Lack of delirium-related knowledge and training among nursing staff	Multiple education and information sessions to educate and train frontline staff
3	Lack of preoperative baseline risk assessment	Preoperative delirium risk assessment (CPAC)
4	Over sedation	Screening patients’ sedation status using the validated RASS scale [14]
5	Delirium screening	Screening for delirium by the RN and Physician Assistant using validated CAM-ICU instrument [15]
6	Perceived pain and discomfort screening	Assessed pain using validated COPT scale [13]
7	Early mobilization	Obtained dedicated mobilization staff (physiotherapist) and trained them for screening patients’ stability, adjusting mechanical ventilation, securing devices and untangling of lines and tubes Providing mobilization-enabled ICU equipment
8	Interventions aimed at preventing delirium	A positive CAM indicates delirium a medical emergency and was always converted to action including investigation and intervention
9	Lack of patient and caregiver engagement	Delirium brochure for patient and caregiver education Collecting patient personal information (such as preferred name, use of hearing/viewing aid) to aid postoperative care provider to understand and provide for patient preferences and communicate effectively. A “getting to know you “form was introduced
10	Lack of communication with the community physician (family physician)	It is essential to provide additional care and support to the patients during their transition in and out of community The family physician was informed if the patients were found at risk of delirium during preoperative assessment as well as if the patient developed delirium during the postoperative period

**Table 5 geriatrics-06-00111-t005:** Process and Outcome Measures.

Measure (S)	Mode of Assessment
**Process Measures (Intervention adherence)**	
Intervention adherence	Rate of completion of baseline risk assessment
	Rate of completion of delirium assessment
**Outcome Measures**	
Primary outcome	Rates of delirium screening
Quality indicators	Number of positive CAM screens in clinical database for patients screened with CAM
	Number of patients restrained in ICU, Wards
	Hospital LOS, ICU LOS
	All-cause in hospital mortality
	Major adverse cardiac events
	Rate of sternal wound infection

**Table 6 geriatrics-06-00111-t006:** Patient Characteristics During Delirium Quality Improvement Period.

Patient Characteristic	Pre-rPOD Intervention (2009–2011)	During-rPOD Intervention (2012–2015)	Post-rPOD Intervention (2016–2018)	*p*-Value
Age	66 (58–74)	67 (58–74)	68 (59–75)	<0.001
Sex (Female)	28.2%	27.9%	27.8%	0.942
Type of Cardiac Surgery				
CABG	60.9%	46.9%	45.6%	<0.001
Valve	14.8%	20.2%	23.0%	<0.001
CABG + Valve	11.2%	11.3%	11.1%	0.969
Other	13.1%	21.6%	20.3%	<0.001
MoCA Score	-	26 (23–28)	26 (23–28)	0.479
CFS (Nursing Assessment)	-	3 (2–4)	4 (3–4)	<0.001
Patient Health Questionnaire (Version 9)	-	1 (0–3)	2 (0–6)	<0.001

Continuous variables expressed as median (quartile 1–3) and compared using Kruskal-Wallis Test. Categorical variables expressed as percentage and compared using Chi-Square Test. Summary statistics calculated on non-missing data. MoCA—Montreal Cognitive Assessment, CFS—Clinical Frailty Scale

**Table 7 geriatrics-06-00111-t007:** Outcome Data and Compliance—Process Measures.

7.1. Process Measures	During-rPOD Implementation (2012–2016)	Post-rPOD Implementation (2016–2018)	*p*-Value
**7.1.1 Baseline risk assessment**			
MoCA completion rate	30.0%	46.6%	<0.001
CFS completion rate	30.7%	49.2%	<0.001
**7.1.2 Delirium assessment**			
Any CAM Assessment Recorded	97.3%	98.5%	0.002
Any RASS Assessment Recorded	98.4%	99.2%	0.006

Continuous variables expressed as median (quartile 1–3) and compared using Kruskal-Wallis Test. Categorical variables expressed as percentage and compared using Chi-Square Test. Summary statistics calculated on non-missing data. MoCA—Montreal Cognitive Assessment, CFS—Clinical Frailty Scale.

**Table 8 geriatrics-06-00111-t008:** Outcome Data and Compliance—Quality Improvement Measures.

7.2. Quality Improvement Measures	Pre-rPOD Intervention (2009–2011)	During-rPOD Intervention (201–2016)	Post-rPOD Intervention (2016–2018)	*p*-Value
**7.2.1 Primary outcome**				
Postoperative delirium screening rates	9.0%	23.3%	19.1%	<0.001
**7.2.2 Quality indicators**				
Number of Positive CAM screens in clinical database for patients screened with CAM	2 (1–4)	2 (1–5)	3 (1–6)	<0.001
Number of patients restrained—ICU	-	3.0%	1.2%	<0.001
Number of patients restrained—Ward	-	0.4%	0.4%	0.955
Length of ICU stay for patients screened with delirium (Hours)	79 (43–161)	90 (42–165)	74 (41–147)	0.329
Length of Hospital Stay (Surgery to Discharge) for patients screened with delirium (Days)	12 (7–22)	13 (8–23)	12 (8–23)	0.282
Major Adverse Cardiac Events (MI, Stroke, Dialysis, In-Hospital Mortality)	5.4%	8.1%	6.3%	<0.001
Sternal Infection (Superficial or Deep)	0.2%	1.3%	1.3%	<0.001
In-Hospital Mortality	2.5%	3.2%	2.1%	0.012

Continuous variables expressed as median (quartile 1–3) and compared using Kruskal-Wallis Test. Categorical variables expressed as percentage and compared using Chi-Square Test. Summary statistics calculated on non-missing data.

## Data Availability

Not applicable.

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
