# Peer review of "A Quality Improvement Initiative to Reduce Postoperative Delirium among Cardiac Surgery Patients"

_geriatrics, 2021, doi:10.3390/geriatrics6040111_

Round 1
Reviewer 1 Report
Very intereting paper sell sfittente with high scientific content
Author Response
We appreciate your through review and thoughtful comments of our manuscript entitled “A quality improvement initiative to reduce postoperative delirium among cardiac surgery patients.”. We have comprehensively revised our manuscript based on the reviewers’ comments as well as additional edits to improve readability. We have our detailed response as below.
Reviewer #1
- Very interesting paper sell sfittente with high scientific content
Response: Thank you so much for your kind words and your time to review our manuscript.
Reviewer 2 Report
I am pleased to read this manuscript about a very important quality improvement initiative to reduce postoperative delirium. The topic is espccially in the ageing population of major concern.
The authors described the model of implementation very clearly. I find it particularly good that the integration of patients and their relatives into the preventive measures has been taken into account. The concept to implement a bundle of measures to reduce the incidence of delirium is well described in the methods part of the manuscript.
But I am quite a bit confused, when I am reading the results. Perhaps it is my fault and I just do not understand it correct.
Your primary outcome is the following: "The primary outcome measure was the change in rates of POD screening before, during and after the rPOD care bundle implementation." You say the postoperative screening rate is 9,0%, 23,3% and 19,1% before, during and after the intervention. But the delirium was assessed with CAM in 97.3 and 98.5% during and after the implementation. I do not understand, what you measure exactly as primary outcome. Could you please explain this more precise in your manuscript?
Some other issues are:
- How many patients were included into the analysis?
- How did you analyse the care expediture?
- A possible bias could be the shift from CABG to valve operations over time. Please discuss this.
- How do you explain the prolonged ICU stay (79 hours vs. 90 hours) before and during the intervention?
- The abstract should contain some of the results not only the methods.
Author Response
We appreciate your through review and thoughtful comments of our manuscript entitled “A quality improvement initiative to reduce postoperative delirium among cardiac surgery patients.”. We have comprehensively revised our manuscript based on the reviewers’ comments as well as additional edits to improve readability. We have our detailed response as below.
Reviewer #2
- I am pleased to read this manuscript about a very important quality improvement initiative to reduce postoperative delirium. The topic is especially in the ageing population of major concern. The authors described the model of implementation very clearly. I find it particularly good that the integration of patients and their relatives into the preventive measures has been taken into account. The concept to implement a bundle of measures to reduce the incidence of delirium is well described in the methods part of the manuscript.
Response: Thank so much dear reviewer for taking time to review our manuscript. We really appreciate your kind words.
- But I am quite a bit confused, when I am reading the results. Perhaps it is my fault and I just do not understand it correct. Your primary outcome is the following: "The primary outcome measure was the change in rates of POD screening before, during and after the rPOD care bundle implementation." You say the postoperative screening rate is 9,0%, 23,3% and 19,1% before, during and after the intervention. But the delirium was assessed with CAM in 97.3 and 98.5% during and after the implementation. I do not understand, what you measure exactly as primary outcome. Could you please explain this more precise in your manuscript?
Response: Thank you so much for your feedback. As rightly stated, the primary outcome measure was the rate of postoperative delirium screening. To calculated this variable, we included only those CAM screening test data which were complete. i.e. all the screening questions were answered. All incomplete data was excluded for evaluating this variable. To clarify we have included additional text in line 261 of the manuscript. On the other hand, ‘Any CAM assessment’ variable was calculated using all the available data, irrespective of whether complete CAM assessment tool was administered or not. We had included data even if CAM assessment was initiated but not completed. Relevant text has been included in line 234 of the manuscript.
Some other issues are: Thank you so much for bringing the below questions to our attention. We have included the relevant data in the manuscript as described below.
- How many patients were included into the analysis?
Response: In total 3340 patients were included for analysis. We have included the data in line 230
- How did you analyze the care expenditure?
Response: The healthcare expenditure data was collected from the administrative database and was stratified based on the study period. We have added relevant text in the cation of Figure 2.
- A possible bias could be the shift from CABG to valve operations over time. Please discuss this.
Response: Thank you so much for the suggestion. We have included this description in the limitation section.
- How do you explain the prolonged ICU stay (79 hours vs. 90 hours) before and during the intervention?
Response: Thank you so much for this important question. The ICU length of stay data demonstrated a shorter ICU stay before and a longer stay during rPOD intervention implementation (as stated above). This could be attributed to insufficient data points (small sample size) for the period before the implementation of rPOD intervention. From the statistical perspective as well, the confidence intervals for length of ICU stay before and during the implementation of rPOD intervention were found to be overlapping. This re-iterated that there was no difference between the median ICU length of stay before and during the rPOD intervention implementation. We had included this data for completeness and transparency for the readers.
- The abstract should contain some of the results not only the methods.
Response: Thank you so much for the suggestion. We have updated the manuscript with relevant data.
Reviewer 3 Report
In the paper "A quality improvement initiative to reduce postoperative delirium among cardiac surgery patients" the authors describe the implementation of a quality improvement program aimed to reduce postoperative delirium. The paper is well presented and describes the phases of implementation as well as measures of monitorization of this program. The limitations are also acknowledged. The manuscript is an important contribution for the development and implementation of delirium improvement programs.
Author Response
We appreciate your through review and thoughtful comments of our manuscript entitled “A quality improvement initiative to reduce postoperative delirium among cardiac surgery patients.”. We have comprehensively revised our manuscript based on the reviewers’ comments as well as additional edits to improve readability. We have our detailed response as below.
Reviewer #3
Comments and Suggestions for Authors
- In the paper "A quality improvement initiative to reduce postoperative delirium among cardiac surgery patients" the authors describe the implementation of a quality improvement program aimed to reduce postoperative delirium. The paper is well presented and describes the phases of implementation as well as measures of monitorization of this program. The limitations are also acknowledged. The manuscript is an important contribution for the development and implementation of delirium improvement programs.
Response: Thank you so much for your appreciation of our research and of our manuscript.
Round 2
Reviewer 2 Report
Dear Authors,
manuscript is fine now, all neccessary information was added.
Nice work!